# Co-Authorship Networks Analysis to Discover Collaboration Patterns among Italian Researchers

**Vincenza Carchiolo** *[ID], **Marco Grassia** [ID], **Michele Malgeri** [ID] and **Giuseppe Mangioni** [ID]

Dipartimento di Ingegneria Elettrica Elettronica Informatica, Università di Catania, Viale Andrea Doria, 9-95127 Catania, Italy; marco.grassia@unict.it (M.G.); michele.malgeri@unict.it (M.M.); giuseppe.mangioni@unict.it (G.M.)

*   Correspondence: vincenza.carchiolo@unict.it; Tel.: +39-095-7382359

**Abstract:** The study of the behaviors of large community of researchers and what correlations exist between their environment, such as grouping rules by law or specific institution policies, and their performance is an important topic since it affects the metrics used to evaluate the quality of the research. Moreover, in several countries, such as Italy, these metrics are also used to define the recruitment and funding policies. To effectively study these topics, we created a procedure that allow us to craft a large dataset of Italian Academic researchers, having the most important performance indices together with co-authorships information, mixing data extracted from the official list of academic researchers provided by Italian Ministry of University and Research and the Elsevier's Scopus database. In this paper, we discuss our approach to automate the process of correct association of profiles and the mapping of publications reducing the use of computational resources. We also present the characteristics of four datasets related to specific research fields defined by the Italian Ministry of University and Research used to group the Italian researchers. Then, we present several examples of how the information extracted from these datasets can help to achieve a better understanding of the dynamics influencing scientist performances.

**Keywords:** network sciences; social network; coauthorship networks

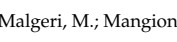



## 1. Introduction

In the last years, the use of bibliometrics [1] gained greater and greater importance in the evaluation of researchers and scientific projects, and it is currently used in several countries (such as Italy) to rank researches and universities, and the attempts at performing this kind of quantitative analysis have often been referred to as "science of science" [2–4]. However, the use of citation indexes is quite old as witnessed by pioneering works of Garfield [5] in the 1950s, and de Solla Price [6]. Today, there are several metrics that tries to quantify the quality and impact of researchers [7–10]. On the other hand, co-authorship networks, i.e., networks where nodes represent scientists and a link between any couple of nodes means the corresponding researchers have co-authored at least a paper, have been extensively studied with the aim of understanding the collaboration patterns among scientists. To approach such a task there are many challenging issues as, for instance, data collection and filtering, definition of domain-specific metrics and algorithms, data visualization, and mining. Co-authorship networks have been widely explored from the perspective of complex networks [11,12], since such representation allows discovering structural and dynamic patterns of scientific collaborations, often hidden, or neglected in the bibliometric approach. Barabási et al. [13] and Newman [14,15] examined the "small-world" and "scale-free" features in the co-authorship networks using the datasets in the disciplines of mathematics, neuro-science, physics, biomedical studies, and computer science. Several works followed the path of these pioneering papers deepening the "scale-free" and "small-world" characteristics [16–18] or investigating the mechanisms for the evolution of the co-authorship networks [19–21].

The enrichment of co-authorship networks with bibliometrics indices allow studying the characteristics and dynamics of such complex networks. In this work, we discuss why and how we created a co-authorship dataset from the Italian academic context. In particular, each academic working in an Italian university belongs to an *academic discipline* (i.e., a grouping by topic imposed by law), which is also used for evaluation and career advancement, and we believe that this fact has a strong impact on both bibliometrics and how academic collaborations arise. Here, we selected four academic disciplines (Mathematical Analysis—MAT/05, Economics—SECS-P/01, Information Processing Systems—ING-INF/05, and Informatics—INF/01) that have almost the same number of researchers but cover different research topics. We built the dataset starting from the profile of Italian researchers and obtaining all the available bibliometrics querying the Scopus database, and we extracted all the bibliometrics up to end of 2021. This paper presents the details of the datasets crafting, such as mapping, solving ambiguity, etc., some preliminary analysis and, finally, a discussion about their characteristics.

The next section briefly summarizes the state-of-the-art, Section 3 deals with method and materials used to create datasets, while Section 4 presents the above-mentioned example and conclusions reports about results, open problems, and further development.

## 2. Related Works

Co-authorship networks have been of great interest for several decades [11,12] and have been one of the main topics in network science research. For instance, in the pioneering work [15], Newman constructed some networks in which nodes are scientists (extracted from bibliographic databases in biology, physics, and mathematics) and two nodes are connected by an edge if they have co-authored almost a paper. Such network were used to study collaboration patterns over time, and how they vary between subjects. Newman highlighted that *"the co-authorship network is as much a network depicting academic society as it is a network depicting the structure of our knowledge. Additionally, perhaps because of this, it has received far less attention than have citation networks"*. In [22] the authors presented a study showing how some topological features impact on some bibliographic indices using networks that encompass only a subset of Italian scientists.

In addition to the works already mentioned in the introduction that mainly focused on network topological properties, such as the "scale-free" and "small-world" properties, other authors used the co-authorship networks to evaluate the performance of a set of researchers. Hâncean et al. [23] exploited the co-authorship networks among the most productive European researchers, over a 12-year time window, looking at the impact of collaboration upon the citations aiming at discovering the best European researchers. Weihua et al. [1] examined the impact of early co-authorship on the careers of junior researchers in four specifics scientific. In [24], the authors explored the correlation between centrality metrics in co-authorship networks and Hirsch index [8] (H-index). In [25], the level of collaboration is evaluated through the definition of a specific centrality index called $\phi$. Other authors examined the evolution over time of the co-authorship networks. For example, Parish et al. [26] studied the dynamics of productivity in different fields of scientific research while Xie [27] proposed a hyper-graph model for simulating the evolution of large co-authorship networks.

Of course, whatever is the goal of the study the starting point is always the choice of scientists to study and the construction of the network itself. It is straightforward that the number of scientists included, but also the set of features, the time interval and several other elements could influence the final results, therefore the method and sources selected are fundamental by themselves to guarantee meaningful results. The main contribute of this paper is the description of the strategies used to construct a dataset from which derive the co-authorship networks enriched by several well-established performance parameters. We also explain issues met during the setup, such as the presence of ambiguous or duplicated information, how to manage missing information, how to combine results and so on.

The source of bibliometrics information is the Elsevier's Scopus Database [28], while the scientists used as seeds was the Italian researchers belonging to official academic organizations. Although there are many other bibliographic databases available, such as Web Of Science (WoS) [29] and Google Scholar [30], we decide to use the Scopus Database since it is widely recognized in Italy, contains almost all publication types and is used as a reference for many Italian academic ranking by law. However, in literature can be found several examples where co-authorship networks are based on different data sources, such as [31] that extracts data from WoS.

Among the recent co-authorship networking available in the literature based on Scopus, Di Bella et al. [32] proposed a temporal analysis of the co-authorship network of Italian Institute of Technologies, and Pradhan et al. [33] evaluated the performance of some universities; both selected the authors (nodes of the network) according the authors' affiliation. Fujita and Vitevitch, in [34], quantified the extent to which Psychology is multidisciplinary, and how it changed over time using the tool provided by network science. They studied the citation network from all the articles published in journals identified by the Web of Science as Multidisciplinary-Psychology for each year from 2008 to 2018.

In our proposal the selection criteria are based on a classification, performed in Italy by law, that groups researchers belonging to public universities according to topics similarity called "Settori Scientifici Disciplinari" that can be translated in English in academic disciplines or specializations. The peculiarity of the proposed dataset is the extension of the network to direct collaborator and collaborators of collaborators outside the authors used as seed, since we believe that topological properties strongly depend on the full network. To the best of our knowledge, such kind of study has not been carried out so far.

## 3. Data Collection

The collection of meaningful and useful data is a task with many challenges. When dealing with collaborations among researchers, the key issues come from the multiple data sources and the data not being structured, which translates, for instance, into ambiguous names or duplicate author profiles. In this section, we introduce the data sources and the data-collection methodology used to build the collaboration networks.

### 3.1. Data Sources

The sources of data used to create the dataset are: the list of researchers employed in Italian public universities provided by Italian Ministry of University and Research (MUR) [35] and the well-known Elsevier's Scopus database. Data were collected at the end of 2021 and contains all records presents in Scopus and Ministry's list at the moment. In particular, the former—which is updated daily and provides the profile of the researchers that are employed in the Italian academia—are used as seed data, and the latter are used to retrieve all the researchers' publications and bibliometrics. As shown in Figure 1, reporting a slice of the data, the information is partially structured, all fields are in Italian language and some of them—such as department or structure—are in free form that do not allow to directly use them to query Scopus. To build a coherent dataset, we build a network starting from the (seed) researchers that belong to a given academic discipline, i.e., the group of people that belong to the research area, as defined by the Italian Law. Specifically, we selected the academic researchers belonging to MAT/05 "Mathematical Analysis" ("Analisi matematica"), SECS-P/01 "Economics" ("Economia politica"), ING-INF/05 "Information Processing Systems" ("Sistemi di elaborazione delle informazioni"), and INF/01 "Informatics" ("Informatica"). We selected these academic disciplines because they have a similar number of academics that allows performing a thorough analysis in a reasonable amount of time, they are all *scientific disciplines* and share some research topics, but are significantly different from each other. Moreover, the authors of this paper belong to one of them (ING-INF/05) and, thus, are aware of some of its dynamics, which can help in

the analysis of the academics' habits. Of course, other academic disciplines share these characteristics and would allow to extend the dataset in the future.

| Fascia | Cognome e Nome | Genere | Ateneo | Facoltà | S.C. | Struttura di afferenza | Servizio prestato in altro ateneo |
|---|---|---|---|---|---|---|---|
| ■ Associato | | M | Scuola Superiore Sant'Anna | | 09/H1 | Istituto di Tecnologie della Comunicazione, dell'Informazione e della Percezione (TECIP) | |
| ■ Ordinario | | M | BOLOGNA | | 09/H1 | Ingegneria dell'Energia Elettrica e dell'Informazione "Guglielmo Marconi" | |
| ■ Ordinario | | M | GENOVA | | 09/H1 | Informatica, bioingegneria,robotica e ingegneria dei sistemi (DIBRIS) | |
| ■ Associato | | M | Politecnico di MILANO | | 09/H1 | Elettronica, Informazione e Bioingegneria | |
| ■ Associato confermato | | F | PAVIA | | 09/H1 | INGEGNERIA INDUSTRIALE E DELL'INFORMAZIONE | |
| ■ Associato | | M | "Ca' Foscari" VENEZIA | | 09/H1 | Scienze Ambientali, Informatica e Statistica | |
| ■ Associato | | M | PARMA | | 09/H1 | Ingegneria e Architettura | |
| ■ Ordinario | | M | Politecnico di MILANO | | 09/H1 | Elettronica, Informazione e Bioingegneria | |
| ■ Associato | | F | Napoli Federico II | | 09/H1 | Ingegneria Elettrica e delle Tecnologie dell'Informazione | |

**Figure 1.** Excerpt of data as provided by MUR. Column labels are role, family and given name, gender, university, school, sector, department, or structure, and note about service.

The Elsevier's Scopus database, according to their fact sheet [28], holds more than 1.8 billion of cited references dating back to 1970, more than 17.6 million of author profiles, and more than 84 million of records. We queried the database using the provided APIs, which expose curated abstracts and citation data from all journals indexed by Scopus, in agreement to their policies [36,37].

*3.2. Mapping and Ambiguity*

Once chosen the academic disciplines and collected information from both sources we must combine them into a unique dataset that contains the information related to academic discipline and all bibliometrics. Unfortunately, no unique identifier, such as OrcID, is present in Ministry's list therefore the ambiguity must be resolved using the information related to affiliation (columns *Ateneo*, the University's name, and *Struttura di appartenenza*, the department). Moreover, the Scopus database is built from the information present in published papers and has multiple profiles for the same author (often already linked), homonyms and partial registrations (researchers registered with partial names or omitted information), and each of those issues can lead to the wrong network representation.

The Scopus Search API is organized into three clusters: Affiliation that has Affiliation Profile, Author that contains Author Profiles, and Scopus contains the abstracts and relevant metadata. Searching against Author cluster retrieves the following attributes useful to uniquely identify the author: "dc:identifier" that is the Scopus unique identifier associated with profile, "OrcID" if any, "surname", "given-name", and "affiliation" split into name, city and country. Despite the details of information given by the query, there are still some problems in mapping the data from the MUR's list to a unique Scopus profile. We report the most important ones in the following:

1. The surname and given-name in the Ministry list is always the legal name while the corresponding fields in Scopus may be abbreviated, misspelled, alias, reverted, and incomplete. For instance, Ministry List refer to "Michele Giuseppe Malgeri" while Scopus profile only contains "Michele Malgeri", some authors have more than one profile that often are not combined, sometimes surname and given-name are inverted, and, of course, several authors with same (or similar) names may exist.
2. The Ministry list affiliation reports the official name of the institution, meaning that the name is always in Italian, while Scopus often contains the abbreviations, acronyms, or the English translation of the University's name. Of course, again, misspelling and typos may be always present.
3. Spaces, stippling, national characters (e.g., the use of accented vocals).
4. Authors during their activity may change affiliation several times, therefore multiple profile could be present.

To face the problems connected with misspelling, stippling, and typos, before the processing each name is cleaned removing all punctuation, unnecessary spaces, and mapping

national characters, if any, to a normalized form. Comparisons are performed using the Levenshtein distance [38]. The *prepare_queries* setups the set of queries using the author's surname and the shuffle of the given name (if more than one) and the initials of the name, for instance "Malgeri Michele Giuseppe", where the surname is "Malgeri", generates the queries listed in Listing 1:

**Listing 1.** List of queries generates by Malgeri Michele Giuseppe. If they fail third condition about affiliation is discarded and queries will be repeated.

```
AUTHLAST(Malgeri) and AUTHFIRST(M) and AFFIL(Università di Catania)
AUTHLAST(Malgeri) and AUTHFIRST(MG) and AFFIL(Università di Catania)
AUTHLAST(Malgeri) and AUTHFIRST(Michele) and AFFIL(Università di Catania)
AUTHLAST(Malgeri) and AUTHFIRST(MicheleG) and AFFIL(Università di Catania)
AUTHLAST(Malgeri) and AUTHFIRST(MicheleGiuseppe) and AFFIL(Università di Catania)
AUTHLAST(Malgeri) and AUTHFIRST(Giuseppe) and AFFIL(Università di Catania)
AUTHLAST(Malgeri) and AUTHFIRST(GiuseppeMichele) and AFFIL(Università di Catania)
```

We describe the algorithm used to search for the Scopus profile of Italian researchers from the Ministry's list in Algorithm 1. Please note that, for sake of simplicity, the algorithm does not include the use of the researcher's affiliation and the validation of data, which is performed when few matches are found.

**Algorithm 1** Searching for a matching profile between Ministry's list and Scopus Authors' cluster.

> **procedure** *search_for_author*(profile)
>     *matches* $\leftarrow$ []
>     *queries* $\leftarrow$ *prepare_queries*(*profile*)
>     **for all** *query* $\in$ *queries* **do**
>         *result* $\leftarrow$ *search_against_Author_Cluster*(*query_string*)
>         **if** *surname* $==$ *result.surname* **then**
>             *result.distance* $\leftarrow$ *Levenshtein_distance*(*profile.given_name*, *result.given$_n$ame*)
>             *append*(*match*, *result*)
>         **end if**
>     **end for**
>     **if** *match* $\neq \varnothing$ *and min*(*distance*) $<$ *threshold* **then**
>         *return*(*Found*)
>     **else**
>         *return*(*Failed*)
>     **end if**
> **end procedure**

Table 1 reports the dimension of data from Ministry's list and the relative consolidated data. Let us note that, by law, a researcher, at a given time, must belong to one academic discipline, therefore, we use that value from Ministry's list. The last column reports the percentage of authors that could not be associated to any Scopus profile either because of failed disambiguation or because the author is not even present in the database. Let us also note that, while it may be possible to recover some of the missed profiles, for instance by manually inserting the researcher's OrcID or Scopus identifier, we simply discard these few records since the error introduced not affect significantly the dataset being created.

**Table 1.** The results of searching and mapping.

| Academic Discipline | Total in the SSD | Matched | % of Missed |
| --- | --- | --- | --- |
| MAT/05 | 723 | 679 | 6.08% |
| SECS-P/01 | 733 | 674 | 7.77% |
| ING-INF/05 | 843 | 785 | 5.75% |
| INF/01 | 1027 | 966 | 6.76% |

*3.3. Making the Coauthorship Network*

After matching and recovering the profile of Italian academic researchers from the MUR's list, the next step is to retrieve the publication list of each of them and build the co-authorship networks. Specifically, we build a network for each of the "academic disciplines" (SSDs) under study, where nodes are researchers and links among them represent collaborations. More in detail, given two nodes $u$ and $v$, the edge $(u, v)$ represents the fact that researchers $u$ and $v$ have co-authored a paper. We enrich the networks with bibliometrics about the researchers (such as H-index, number of published documents, of citations and of co-authors, Years of activity, etc.) from the Scopus' database, and also associate the number of co-authored papers as the edge weight $w_{(u,v)}$.

Although the algorithm used to obtain the publications and build the networks is quite trivial, it must be optimized to reduce the number and the complexity of the queries to avoid filling the query quotas provided by Elsevier and avoid throttling. Please note that such queries were performed against the Scopus cluster instead of Scopus Author because the information about each publication is useful for further processing and verification. Let us also note that, to reduce the number of API calls, we query 25 documents with each call, and we cache result to avoid to repeat a query.

The algorithm, described in Algorithm 2, was executed three times to explore the co-authorship at different "depths", that is, the maximum number of hops from the seed nodes to explore:

1. Depth-zero: that is the co-authorship of Italian researchers only, i.e., it contains only vertices that belong to Ministry's list.
2. Depth-one: in this step we added only authors that have direct connections with Italian researchers.
3. Depth-two: starting from deep-two we also included the co-authors of co-authors ('two' means that we stopped the deep first search at the second step).

Of course, depth-two network contains depth-one that, in turn, contains depth-zero.

---

**Algorithm 2** Deep first search of common publications.

---

**procedure** *Create_Coauthorship*(author_list)  
    *vertices* ← [ ]  
    *edges* ← [ ]  
    **for all** *author* ∈ *author_list* **do**  
        **if** *author* ∉ *vertices* **then**  
            *append*(*vertices*, *author*)  
        **end if**  
        *result* ← *search_against_Scopus_Cluster*(*authors*)     ▷ search for all author's documents  
        **for all** *document* ∈ *result* **do**  
            **for all** *coauthor* ∈ *extract_authors*(*result*) **do**     ▷ extracts list of authors from result  
                **if** (*author*, *coauthor*) ∉ *edges* **then**  
                    *append*(*edges*, (*author*, *coauthor*))  
                    *weight*((*author*, *coauthor*)) ← 1  
                    *append*(*document*, *setof_documents*((*author*, *coauthor*)))  
                **else if** *document* ∉ *setof_documents*((*author*, *coauthor*)) **then**  
                    *weight*((*author*, *coauthor*)) = *weight*((*author*, *coauthor*)) + 1)  
                **end if**  
            **end for**  
        **end for**  
    **end for**  
**end procedure**

---

## 4. Inside the Data

In this section, we briefly analyze the topology of 12 networks (built as described in the previous section) and the bibliometrics of the authors, and highlight some characteristics of the collected samples.

### 4.1. Structure of the Networks

As detailed in the previous sections, we have built the co-authorship networks using the list of Italian academic researchers from four different "academic disciplines" as seeds: for each discipline, we build three subnetworks where we include all the researchers separated by up to one, two, or three degrees of collaborations. As shown in Table 2, reporting the number of nodes and edges of the three datasets, the networks built at different depths have different size. This is expected, since the networks may also include researchers from other institutions or countries. We also extract the Giant Connected Component (GCC, also known as Largest Connected Component, LCC) of the networks, i.e., the largest cluster of connected nodes, capturing the largest group of researchers that have a finite degree of separation from each other. The sizes of such components are shown in Table 3.

**Table 2.** Italian co-authorship networks size of the academic disciplines under analysis.

| Academic Discipline | Depth-Zero | | Depth-One | | Depth-Two | |
|---|---|---|---|---|---|---|
| | Vertices | Edges | Vertices | Edges | Vertices | Edges |
| MAT/05 | 679 | 1990 | 7891 | 13,764 | 338,475 | 1,042,750 |
| SECS-P/01 | 674 | 784 | 10,736 | 13,404 | 727,060 | 2,512,882 |
| ING-INF/05 | 784 | 3759 | 46,879 | 84,127 | 1,900,424 | 6,897,383 |
| INF/01 | 966 | 4327 | 43,339 | 78,283 | 1,824,553 | 6,666,596 |

**Table 3.** Giant Connected Components size of the Italian co-authorship networks of the academic disciplines under analysis.

| Academic Discipline | Depth-Zero | | Depth-One | | Depth-Two | |
|---|---|---|---|---|---|---|
| | Vertices | Edges | Vertices | Edges | Vertices | Edges |
| MAT/05 | 613 | 1970 | 7712 | 13,599 | 338,462 | 1,042,745 |
| SECS-P/01 | 458 | 752 | 9309 | 12,042 | 727,028 | 2,512,863 |
| ING-INF/05 | 758 | 3758 | 46,851 | 84,101 | 1,900,413 | 6,897,354 |
| INF/01 | 929 | 4318 | 43,324 | 78,271 | 1,824,552 | 6,666,596 |

We also plot the GCC of the depth-one networks (for the sake of readability, as the larger networks would be unintelligible) in Figure 2 that highlights the Italian researchers. Let us note that the co-authorship networks of ING-INF/05 and INF/01 disciplines show similar collaboration patterns, which is expected considering the large overlap in research interests and topics between them.

In Table 4, we report some topological characteristics of the depth-two networks, describing their size, edge density, and connection patterns. As the Reader may notice, although the computer science-related fields have comparable size, the networks show a very different number of nodes and edges among academic fields. On the other hand, the average degree, strength, and link weight are comparable. Another interesting difference among the disciplines is in the average local clustering coefficient and in the transitivity, which shows that the researchers in computer science-related fields are more prone to collaborate. We also compute the degree and strength assortativity, as defined in [39,40].

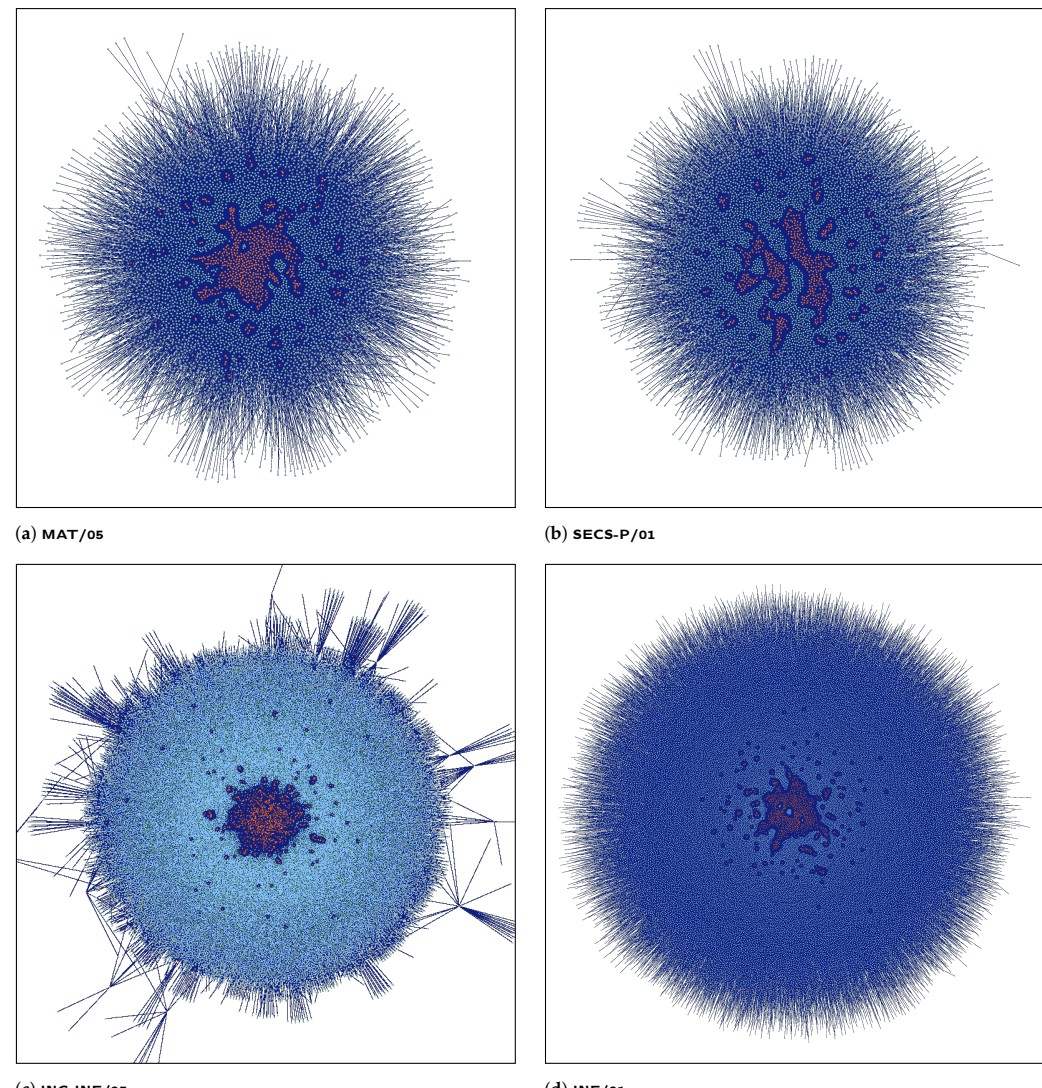

(**a**) MAT/05

(**b**) SECS-P/01

(**c**) ING-INF/05

(**d**) INF/01

**Figure 2.** These figures highlight the core of co-authorship networks (orange/clear) with respect to direct coauthors (blue/darker).

**Table 4.** Characteristics of the Largest Connected Component of the Depth 1 networks.

|  | SECS-P/01 | MAT/05 | ING-INF/05 | INF/01 |
|---|---|---|---|---|
| Number of edges | 12,042 | 13,599 | 84,101 | 78,271 |
| Number of nodes | 9309 | 7712 | 46,851 | 43,324 |
| Avg. degree | 2.58717 | 3.52671 | 3.59015 | 3.61329 |
| Density | 0.00028 | 0.00046 | 0.00008 | 0.00008 |
| Avg. link weight | 2.12465 | 2.62593 | 2.91573 | 2.93882 |
| Avg. strength | 5.49683 | 9.26089 | 10.46791 | 10.61878 |
| Avg. local clustering coefficient [11] | 0.08487 | 0.23174 | 0.27827 | 0.26523 |
| Transitivity [41] | 0.00364 | 0.04231 | 0.01994 | 0.02194 |
| Avg. k-core [42,43] number | 0.09904 | 0.30368 | 0.09470 | 0.11534 |
| Max k-core number [42,43] | 3 | 6 | 17 | 8 |
| Degree assortativity [39] | −0.05231 | −0.13266 | −0.12238 | −0.12909 |
| Strength assortativity [40] | −0.02465 | −0.09091 | −0.11980 | −0.01308 |
| Avg. Shortest Path Len. | 6.25596 | 4.44838 | 3.86187 | 3.93791 |
| Diameter | 16 | 16 | 9 | 10 |

### 4.2. Bibliometrics

In this Subsection, we analyze the bibliometrics associated to the authors. Figure 3 represents the depth-0 networks with a node and edge size proportional to the author's H-index and number of coauthored publications, respectively. Such figures allow to appreciate that relevant nodes are often heavily connected, i.e., they are *hubs*, and also the different habits and policies of the academic disciplines is clearly show by this representation: researchers in the SECS-P/01 discipline collaborate with other Italian researchers belonging to the same discipline less than others, which also translates into a larger number of small Connected Components; MAT/05 shows a very small core with a sort of "tail", a chain of collaborations; INF/01 and ING-INF/05 exhibit a large core and a far denser collaboration network.

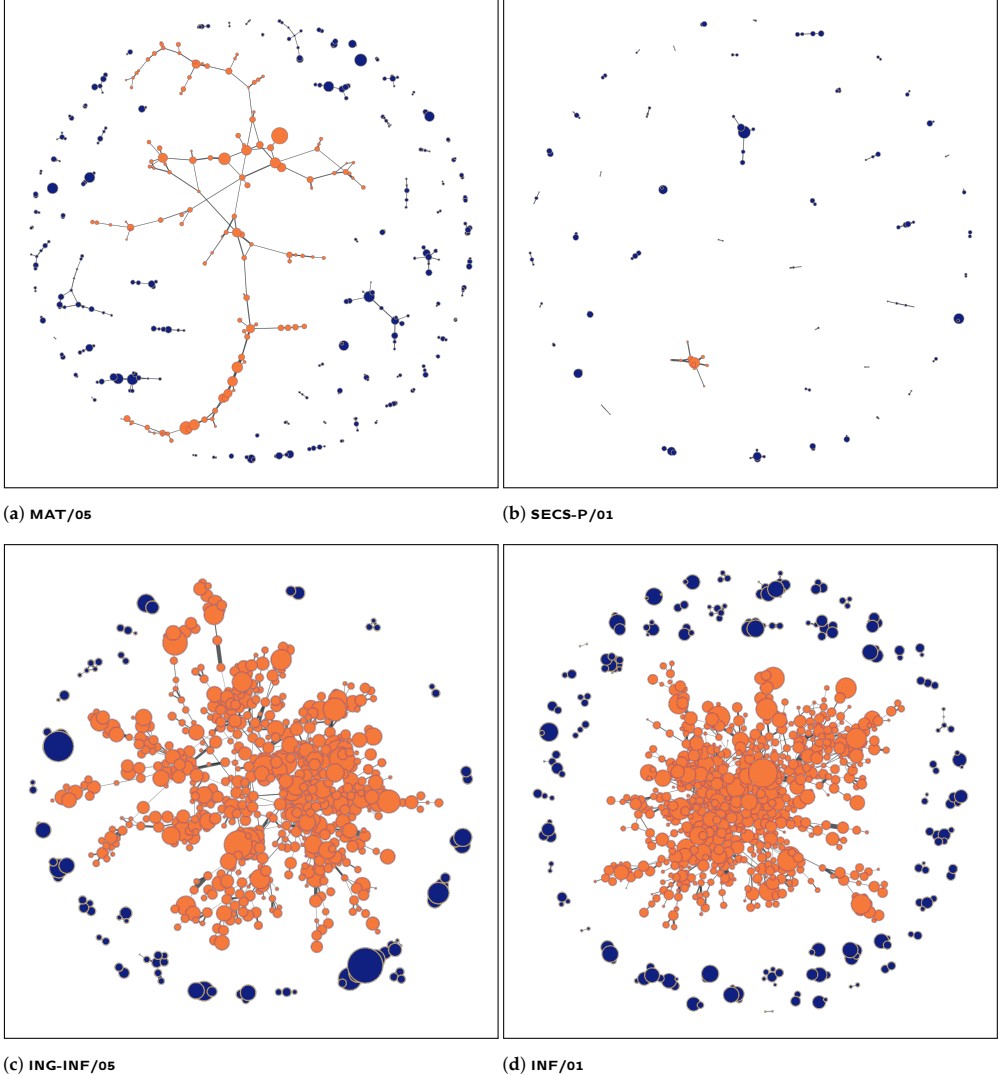

(a) MAT/05              (b) SECS-P/01

(c) ING-INF/05          (d) INF/01

**Figure 3.** Italian academics' co-authorship networks at depth-0. Vertices and edge widths are proportional to the author's H-index edge weights, respectively. The Giant Connected Component colored in orange/clear. For sake of readability, the nodes with degree $d_i < 5$ have been filtered out.

The distributions of the most relevant bibliometrics indices contained in the dataset are shown in Figure 4 with darker color standing for larger density. Again, a strong similarity can be observed between computer science-related fields, while the economics and mathematical analysis disciplines show different distributions. For instance, the H-index range is wider in the ING-INF/05 and INF/01 than the other disciplines, and the distribution is fat-tailed. Regarding the number of documents, of citations and of coauthors

the distributions of INF/01 and SECS-P/01 show a very dense area around small value. On the other hand, number of years activity is similar across all disciplines.

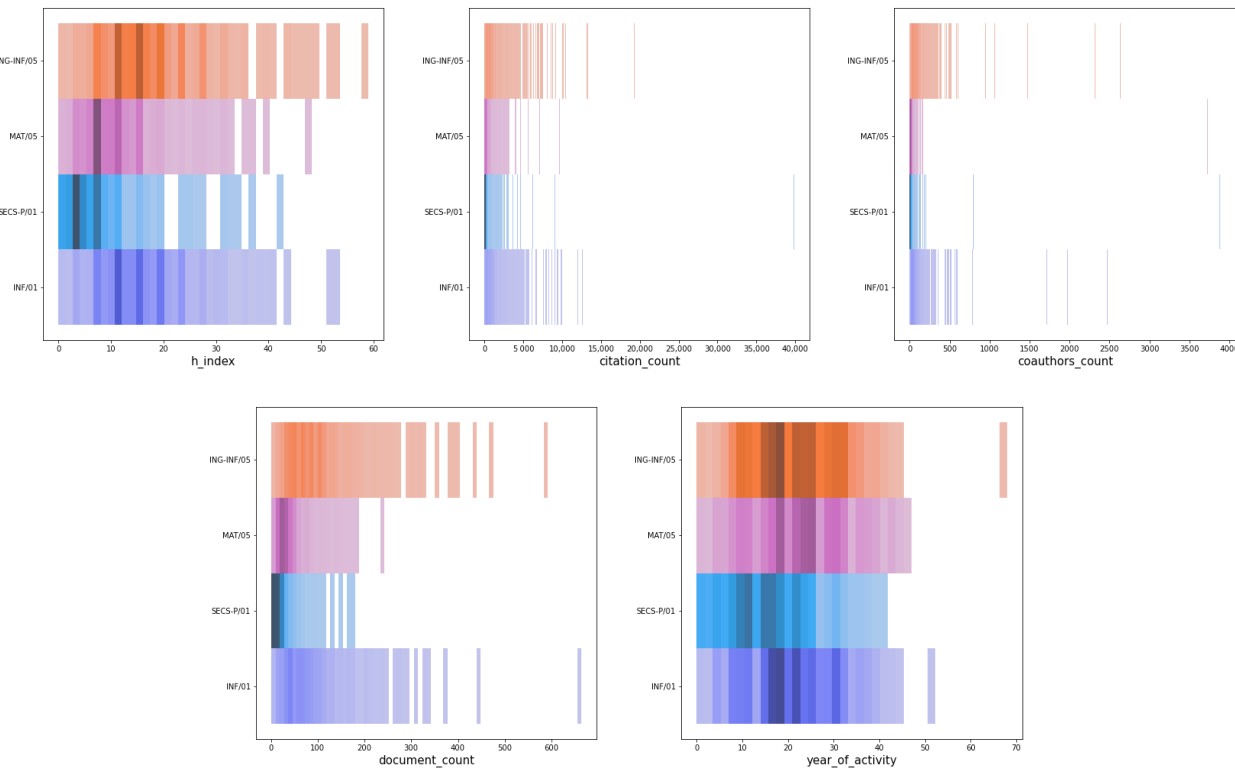

**Figure 4.** Distribution of some bibliometric indices in the academic disciplines.

Another interesting analysis that could be performed on the networks is related to the "Science of Success". Here, we show, in Figure 5, the correlation between H-index and some indices. More specifically, the figure plots the relation between H-index and the other features in the four dataset using a linear model regression and a confidence interval. Table 5 shows the Pearson product-moment correlation coefficient between H-index and the other features in the four dataset. Once again, the table highlights a very strong overlap of ING-INF/05 and INF/01 disciplines for all the bibliometric indices whilst the Pearson's correlation of SECS-P/01 is markedly different.

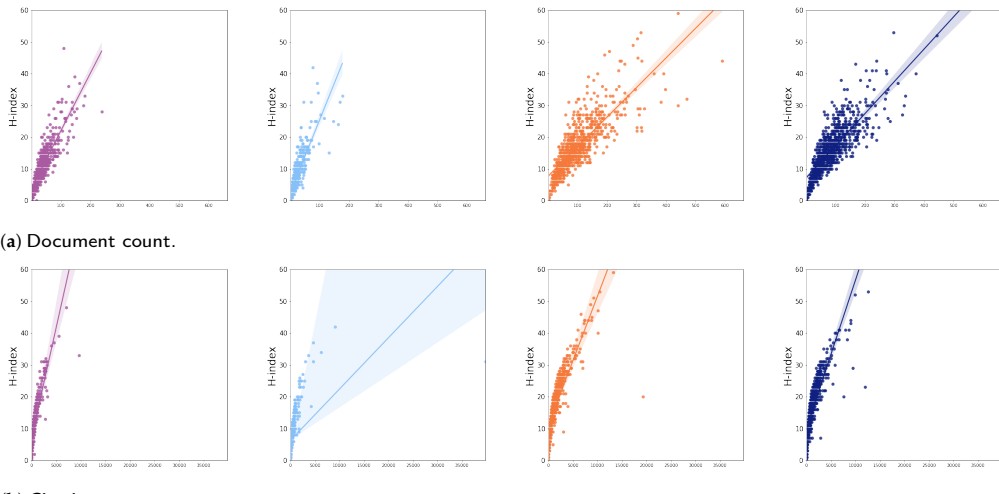

(**a**) Document count.

(**b**) Citation count.

**Figure 5.** *Cont.*

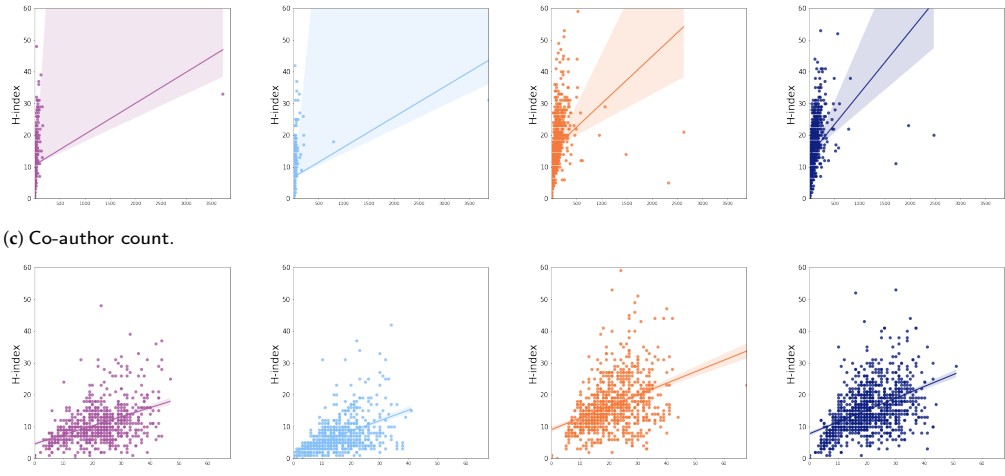

(**c**) Co-author count.

(**d**) Years of activity

**Figure 5.** Correlation among H-index and bibliometric indices in the academic disciplines, from left to right academic disciplines are: MAT/05, SECS-P/01, ING-INF/05, and INF/01.

**Table 5.** Pearson's correlation among H-index and bibliometric indices in the academic disciplines.

| Bibliometric Indices | ING-INF/05 | MAT/05 | SECS-P/01 | INF/01 |
|---|---|---|---|---|
| documents count | 0.80 | 0.85 | 0.85 | 0.82 |
| citations count | 0.84 | 0.86 | 0.49 | 0.84 |
| coauthors count | 0.30 | 0.23 | 0.27 | 0.37 |
| Year of activity | 0.38 | 0.43 | 0.49 | 0.44 |

### 4.3. Multidisciplinarity

In this subsection, we aim at answering the following questions: *do researchers belonging to different academic disciplines work together?* and *does the grouping imposed by the Italian law affect the collaboration patterns among researchers?* To do so, we merge the networks of the different disciplines together and show the resulting networks in Figure 6.

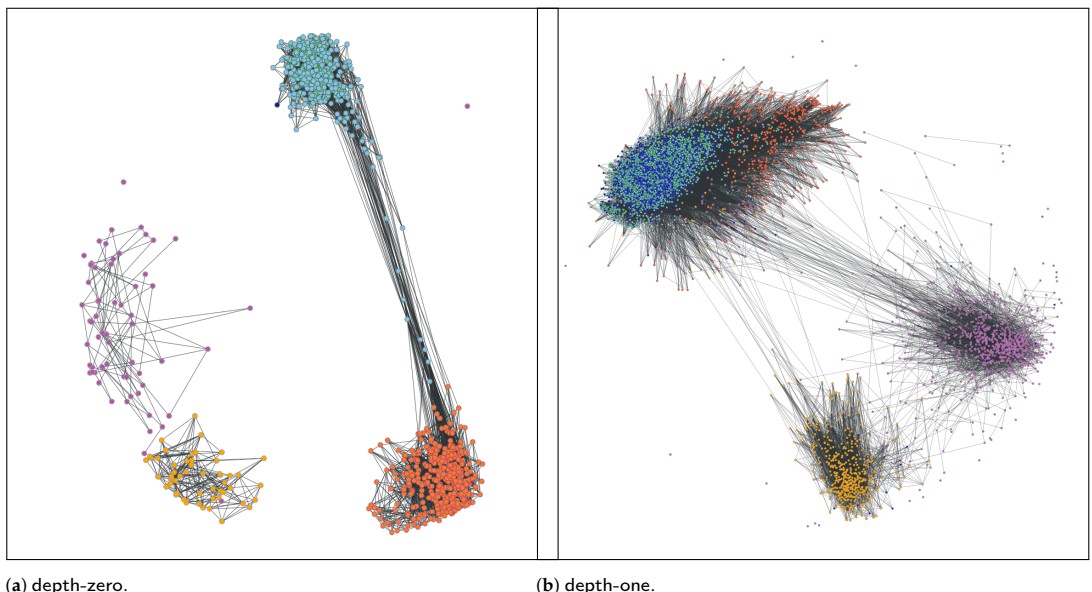

(**a**) depth-zero.                                             (**b**) depth-one.

**Figure 6.** Co-authorship network of the four academic disciplines (ING-INF/05 dark orange, INF/01 cyan, MAT/05 is orange, and SECS-P/01 is violet).

At first, we limit the analysis to the depth-zero networks Figure 6a, which only include Italian academic researchers. As shown in the plot, the number of collaborations across the disciplines under study is very low. More in detail, there is almost no cooperation between researchers in MAT/05, SECS-P/01, and the computer-science related ones, while researchers ING-INF/05 and INF/01 have coauthored more papers, which is expected since the strong overlap in the research topics [44]. We later extend the networks by including the direct coauthors, i.e., by merging the depth-one networks. The resulting network, depicted in Figure 6b, shows that there is some (indirect) collaboration among groups, and a very strong overlap in the groups of INF/01 and ING-INF/05. (Let us note that in depth-1 and depth-2 some authors can be coauthors of researchers belonging to different academic disciplines, so they are present in both networks; we represent them as blue points). By merging the networks at depth-two, the size of the largest cluster grows a lot, which shows that other researchers (e.g., researchers from the same field but working for private institutions or living abroad) bridge the various groups. Please note that we omit the resulting figure since the network would be too large and dense.

## 5. Conclusions

The collaboration patterns among researchers, and also their relation to higher performance metrics are widely investigated problems, with applications, for instance, in the science of success. In this work, we propose and briefly study a new dataset of collaboration among Italian academic researchers. In particular, we first describe the data-collection process, the challenges and the solution to various problems. Then, we build and analyze the collaboration networks, in search of the different characteristics of four different academic disciplines (groups of researchers defined by the Italian law), however we mainly focused on the approach to build the datasets giving some example of the analysis that could be completed leaving explanation of the results to further works. Our results show similar collaboration patterns among researchers of computer science-related fields (ING-INF/05 and INF/01) and, even if with a limited extent, collaborations among them. On the other hand, researchers from Economics (SECS-P/01) and Mathematical analysis (MAT/05), tend to collaborate less with other Italian colleagues.

### 5.1. Discussion

In this section we provide a brief discussion on contribution of this paper that mainly lies in the datasets and their features, the analysis of them have the purpose to inspire further researches aiming at both studying this specific co-authorship with a well established approach and to explore new approaches, such us the use of the network science tools, to understand hidden mechanism that drive the behavior of scientist communities.

The creation of new dataset represents an important contribution itself, however, it must cover a real world case that exploits some new characteristics and creation strategies to add to previous studies. The case described in this paper has the following peculiarities:

- The scientist community is stable and well-defined since it is defined by law;
- The four dataset share some characteristics (scientific/technical topics, publication areas) but have different publication policy (e.g., number of authors, type of contribution);
- The algorithm, as discussed, can be used to easily extract more academic disciplines in a reasonable time;
- Almost all problems dealing with ambiguity and wrong classification has been solved (the error are $\approx 5\%$, of course, the error can be reduced by human involvement, but it does not modify the structure of co-authorship network;

Measuring the performance of scientist using performance indices, such us bibliometrics, is widely debated but, since it is adopted by more and more institution any contribute that study the correlation between network structure and bibliometrics, if any, permit defining more correct policies (such as recruitment policies) or limit abuses (e.g., publishing "salami slices" with similar methods and slightly different results). However,

the exploratory examples of analysis provided in this paper simply aim at validating the datasets and show its effectiveness in highlight specific characteristics.

*5.2. Future Works and Limitations*

Future works may involve enriching and improving the dataset, for instance, by extending the groups analyzed either to other academic disciplines or to more collaboration hops. Moreover, further analysis can be performed, for example, with data science and network science approaches. Finally, an attempt to explain both the internal behaviors and the difference among academic disciplines is a further step in the comprehension of some dynamics and could lead to understand why some people are more successful than others.

The dataset, since covers several years and long-lasting collaborations could be used to explore the trust relationship among scientists and the impacts of aging [45], or strategies and the cost to build strong communities [46].

The dataset proposed in this paper, as well as the proposed analysis, does not include any studies on the content of the co-authored paper and on the different contribution of each author (if any are documented). Some additional features, present in Scopus DB, such as abstracts and keywords using text analytic techniques [47], could allow searching for very similar publication, and, for instance, to evaluate the impact in bibliometrics but also in co-authorship. However, the dataset contains all information useful to enhance and complete the included information since it includes most of the unique key universally adopted (e.g., OrcID, Scopus-id) that can be used to query both Scopus itself and other archives.

**Author Contributions:** The authors contributed equally to each stage of this work. All authors have read and agreed to the published version of the manuscript.

**Funding:** This work has been partially supported by the project of University of Catania PIACERI, **PIA**no di in**CE**ntivi per la **RI**cerca di Ateneo.

**Data Availability Statement:** The networks analyzed in this study were derived from Elsevier's Scopus DB and are available upon request.

**Conflicts of Interest:** The authors declare no conflict of interest.

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
