# Peer review of "Co-Authorship Networks Analysis to Discover Collaboration Patterns among Italian Researchers"

_futureinternet, doi:10.3390/fi14060187_

Round 1
Reviewer 1 Report
The article aims to explore co-authorship networks among Italian researchers in four academic fields. This study has the strength of empirical analysis based on data. However, I think that it should be improved on a number of dimensions, which I briefly point out.
First, the authors took efforts to collect and clean data. It is not easy work but it is a core part of this study. Authors provided algorithms and table 1, but I could not figure out the data they collected over the years and how many papers they collected in this study. The total in table 1 is unclear and could be estimated by number of authors. Also, I wonder whether authors examine the cases of the same name or overlapping fields among four disciplines and how to clean these cases.
Second, figure 1 and 2 are meaningless. I didn’t understand what authors addressed through these figures. However, figure 3 to 6 are interesting and so authors should explain and interpret these results. Current results did not include interpretation but only a fer descriptive explanations.
Author Response
"Please see the attachment."

Reviewer 2 Report
Using citation networks, this study looks at the correlation between the behaviors of large community of researchers and the context in which they are working. The study uses data from the official list of academic researchers provided by Italian Ministry of Education and the Elsevier's Scopus database. The study found that there is a positive correlation between the two.
The paper presents analysis on an innovative dataset of national relevance for Italy. The authors present a highly detailed data mining pipeline and some qualitative network-based analyses that make the work interesting and innovative, in line with the scope of this journal venue. There are some issues that should be addressed before publication and that is why my recommendation is acceptance after moderate revisions.
---
Key points:
Can we say that network construction influences the features observed across different fields and thus makes them all roughly equal? What are the differences? For instance, in Figure 6 (which needs to have labels increased in font size), what are the R^2 of those scatter plots? Are there statistically significant differences between fields that better highlight different collaboration patterns? This aspect has been already highlighted in text but making it stronger would attract more citations in the future.
Please consider also mentioning more relevant citations to this aspect. For instance, could the overlap in the network structure be due to multidisciplinary aspects? A recent study about this in Psych. has been Fujita and Vitevitch, Nat Hum Soc Sci Comm, 2022.
There is ample debate about the issues with settori scientifico disciplinari hampering innovative multidisciplinary research in the Abilitazione Scientifica Nazionale in Italy. Are the results in the current study of any relevance for those?
A limitation that should be discussed in the Conclusions is that the current study considers only co-signing patterns but doesn't employ additional features available in Scopus, like abstracts and keywords. Using text analytic techniques, see Stella, PeerJ Comp Sci, 2020 or Amancio, Plos One, 2015, one could build also co-occurrence networks out of different scientific fields and see if not only co-authoring patterns but also writing patterns are the same, e.g. do authors tend to publish "salami slices" with highly similar methods and slightly different results? This is a perspective that should be mentioned and briefly discussed, spinned as a future direction.
Minor points:
In Figure 2, red nodes are against green nodes, making the visualisation unreadable to colour blind people or when printed in gray. Please consider using different shapes, instead.
Figure 3 seems like a collation of spaghetti monsters. Are these figures insightful? Could they be replaced by a table with statistics over network cores?
Table 4 should be better commented in text. Why are those features relevant?
Figure 5 is barely readable. Please add labels on the Y axis and change green/red to something else.
The English is generally fine but I would recommend having the paper double checked by a native English speaker to perform some final polishing (e.g. not necessary spaces -> unnecessary spaces; where surname is -> where the surname is; the Years of activity -> years of activity; have different size This is expected -> have different size. This is expected; etc.). This is a very minor point.
---
Overall this is a quite innovative work, the authors should be praised for tackling this innovative route in network science and Italian official data.
Author Response
"Please see the attachment."

Reviewer 3 Report
Dear authors,
Thank you for giving me the opportunity to read your paper. It was really an interesting reading. Please see my following comments for improving its quality.
You have mention the MIUR that actually is MUR.
Figure 1 must be proposed in English and as a table.
Please mention in the introduction and the methodology the period of analysis.
Please explain how did your select just few academic sectors and why or explain that is an explorative study
A discussion section needs to be provided. How your paper fit a gap in the literature. Discuss your results in accordance (or disaccordance) with the previous literature.
Author Response
"Please see the attachment."

Round 2
Reviewer 1 Report
Thanks for your effort to revise this manuscript!
Reviewer 2 Report
The authors addressed all my points and their manuscript is suitable for publication.